



# Intensity-Duration-Frequency (IDF) rainfall curves in Senegal

Youssouph Sane[1], Geremy Panthou[2], Ansoumana Bodian[3], Theo Vischel[2], Thierry Lebel[2],
Honore Dacosta[4], Guillaume Quantin[2], Catherine Wilcox[2], Ousmane Ndiaye[1], Aida Diongue-Niang[1],
and Mariane Diop Kane[1]

[1]Agence Nationale de l'Aviation Civile et de la Météorologie (ANACIM), Dakar, Sénégal
[2]Univ. Grenoble Alpes, IRD, CNRS, Grenoble INP, IGE, 38000 Grenoble, France
[3]Laboratoire Leidi, Université Gaston Berger, Saint Louis, Sénégal
[4]Département de Géographie, Université Cheikh Anta Diop, Dakar, Sénégal

*Correspondence to:* G. Panthou (geremy.panthou@univ-grenoble-alpes.fr)

**Abstract.** Urbanization resulting from a sharply increasing demographic pressure and the development of infrastructures have made the populations of many tropical areas more vulnerable to extreme rainfall hazards. Characterizing extreme rainfall distribution in a coherent way in space and time is thus becoming an overarching need that requires using appropriate models of IDF curves. Using 14 series of 5-min rainfall records (aggregated at a basis time-step of 1h) collected in Senegal, a comparison

5  of two GEV&scaling models is carried out, leading to adopt the most parsimonious one, built around four parameters. A bootstrap approach is proposed to compute the uncertainty associated with the estimation of these 4 parameters and of the related rainfall return levels for durations ranging from 1h to 24h. This study confirms previous works showing that simple scaling holds for characterizing the time-space structure of extreme rainfall in tropical regions such as sub-Saharan Africa. It further provides confidence intervals for the parameter estimates, and shows that the uncertainty linked to the estimation of

10  the GEV parameters, is 3 to 4 times larger than the uncertainty linked to the inference of the scaling parameter. From this model, maps of IDF parameters over Senegal are produced, providing a spatial vision of their organization over the country, with a North to South gradient for the location and scale parameters of the GEV. An influence of the distance from the ocean was found for the scaling parameter. It is acknowledged in conclusion that climate change renders the inference of IDF curves sensitive to increasing non stationarity effects, requiring to warn end-users that they should be used with care and discernment.



## 1 Introduction

The fast growing pressure of mankind on planet Earth makes populations increasingly exposed to hydrometeorological hazards such as torrential rains and floods (IPCC, 2012; Mechler and Bouwer, 2015). Hydrologists are thus more compelled than ever to deal with the problem of assessing the probability of extreme rainfall events at different time scales and for various

return periods, depending on the area of the target catchment and the issue at stake most notably human life protection and infrastructure dimensioning. A classical way of synthesizing the results of such studies is the production of so-called rainfall Intensity-Duration-Frequency (IDF) curves, providing estimates of rainfall return levels over a range of durations. In doing so, scientists are facing two sets of difficulties, one related to data availability and the other to the necessity of a proper methodological framework.

On the data side, the frequency analysis of extremes requires long and continuous records of rainfall at the same location, something not unusual at a daily time-step even though in some regions this is not so common. Moreover a complicating factor is that, in many cases, it is necessary to consider sub-daily time-steps, for which long term records of rainfall are much less numerous or much less reliable/accurate than daily series.

The methodological challenge arises from the complex combination of factors causing rainfall to be strongly variable at all

scales (from the microphysics droplet scales to synoptic scales), as a result of the non-linear interaction of different atmospheric processes (e.g. Schertzer and Lovejoy, 1987). This implies that it is not at all obvious to find a proper theoretical framework to compute IDF curves in a way that ensures coherency between time scales. Early works on IDF proposed empirical methods consisting of first adjusting a frequency distribution models fitted to rainfall series $\{R(D)\}$ for each duration $D$ of interest; and then IDF formula $\{i_T(D)\}$ fitted independently to each series of quantiles derived from the first step and corresponding

to a given return period $T$ (see e.g. Miller et al., 1973; NERC, 1975). This has the advantage of being easily implementable and is thus commonly used by hydrological engineers and operational climate/hydrological services. However, because of uncertainties in the computation of the quantiles derived for the different durations, the scaling formulation may be physically inconsistent and may lead to gross errors such as parasitic oscillations or intersections between IDF curves computed for two different durations (see Koutsoyiannis et al., 1998, for more details). As a remedy to such inconsistencies, Koutsoyiannis et al.

(1998) were the first to propose a general IDF formulation that remains consistent with both the foundations of the probabilistic theories and the physical constraints of scaling across durations. Another notable advance was provided by Menabde et al. (1999) who demonstrated that changes in rainfall distribution with duration formulated by Koutsoyiannis et al. (1998) can be expressed as a simple scaling relationship, opening the path for using the fractal framework in order to describe the time scaling between IDF curves established over a range of durations in various regions of the world (see e.g. Yu et al., 2004; Borga et al.,

2005; Gerold and Watkins, 2005; Nhat et al.; Bara et al., 2009; Blanchet et al., 2016; Rodríguez-Solà et al., 2016; Yilmaz et al., 2016).

Disposing of a consistent scaling framework does not suppress, however, the crucial sampling issues associated with the estimation of the parameters of the IDF model. This involves significant uncertainties in the final determination of rainfall return levels, a question rarely addressed in the literature; on that aspect, see the pioneering work of Mélèse et al. (2017) which





present and compare different methods to compute IDF confidence intervals (on a GEV&scaling model) over the Mediterranean region. It is especially important to investigate this issue in tropical regions such as Sub-Saharan Africa, where a number of new infrastructure projects are in the pipe while at the same time a significant increase of flood risks and related man casualties has been reported over the past 15 years or so (Di-Baldassarre et al., 2010; Tschakert et al., 2010; IPCC, 2014). Several recent

studies have dealt with the question of IDF calculation for different West African countries. Some focused on analysing the behaviour of extreme rainfall distribution at a given location (such as Soro et al., 2008, 2010, for Ivory Coast) while others looked at the scaling behaviour over durations such as Mohymont and Demarée (2006) for Congo and Oyegoke and Oyebande (2008) for Nigeria. Van-De-Vyver and Demarée (2010) also analysed the scaling properties of rainfall over a range of durations for a couple of stations in Congo, finding the value of the main scaling parameter to be larger than the one obtained for Uccle in

Belgium, meaning that the small durations are heavily driving the behaviour of extreme rainfall at larger durations for a tropical climate. More recently, Panthou et al. (2014b) and Agbazo et al. (2016) showed that the GEV&simple-scaling framework is well suited to estimate rainfall return levels at various durations in a coherent way for an array of stations covering a mesoscale area of typically a dozen thousands $\mathrm{km}^2$, respectively in South-West Niger and in Northern Benin. While Agbazo et al. (2016) assume a Gumbel distribution of the annual maxima, Panthou et al. (2014b) used the approach in its broader formulation,

showing that the annual maxima distribution was heavy tailed (positive value of the shape parameter of the GEV). It is worth noting that in both cases use was made of the high quality and fine time scale resolution data collected by the AMMA-CATCH research observatory (Lebel et al., 2009). This data set covering homogeneously a wide range of time-steps (from 5 minutes upward) over more than 20 years is unique in Tropical Africa. This means than in every other area, the parameters of the scaling relationship will have to be inferred from a very limited number of sub-daily rainfall series, not all of them being of

same records length, thus raising the question of which parameters have the largest influences on the final uncertainties of rainfall return levels. This issue is extremely important when dealing with large regions (such as a whole country) over which the scaling parameters may vary spatially, making it unstraightforward to infer rainfall return levels for sub-daily durations when only daily data are available. Focusing on Senegal, a region of contrasted coastal to inland semi-arid climate, our paper ambitions are both to address the uncertainty issue not dealt with in above mentioned papers and to provide IDF curves for

a region located at the western edge of the Sahel, looking at the spatial variability generated by the transition from the coast to inland. In addition to its methodological bearing, the paper aims at making these IDF curves widely accessible to a large range of end-users in the whole country by mapping the values of the scaling parameters and of the rainfall return levels. Furthermore, selecting an IDF model as less sensitive as possible to data sampling effects and computing the associated IDF confidence intervals make easier the update of the IDF curves when new data are available.

## 30   2   Data and Region

### 2.1   Senegal Climatological Context

Senegal is located at the western edge of the African continent between latitudes $12°\mathrm{N}$ and $17°\mathrm{N}$ (Figure 1a). The climate of Senegal is governed by the West African monsoon (Lafore et al., 2011; Janicot et al., 2011; Nicholson, 2013), resulting in a





two-season annual cycle: a dry season marked by the predominance of maritime and continental trade winds in winter and a rainy season, marked by the progressive invasion of the West African monsoon (Figure 2a to Figure 2c) during the summer. The length of the rainy season varies along the latitude and ranges roughly from 5 months (early of June to the end of October) in the South to 3 months (mid-July to mid-October) in the Northern part of Senegal. Rainfall amounts peak in August and

September, coinciding with the period when the ITCZ reaches its northernmost position over Senegal.

There is a strong North-South gradient of the mean annual rainfall (Figure 2d) ranging from 300 $mm$ in the North to more than 1000 $mm$ in the South (Diop et al., 2016). This gradient is mainly explained by the number of rainy days (in average between 20 and 80 from North to South) and to a lesser extent by the mean intensity of rainy days (in average between 10 and 15 $mm\ day^{-1}$), see Figure 2e and 2f.

The rains are mainly caused by Mesoscale Convective Systems sweeping the country from east to west (Laurent et al., 1998; Mathon et al., 2002; Diongue et al., 2002). Sometimes it also happens that cyclonic circulations off the Senegalese coast direct moisture laden air flow over the western part of the country, dumping heavy rains that often cause floods in coastal cities. Due to its convective nature, rainfall is strongly variable in space and time, especially at the event scale for which large differences in rainfall amounts are frequently observed in two very close points (Sane et al., 2012). The rain durations are also generally

short, except in the rare case of stationary convective systems (blocked situation).

Senegal regularly undergoes damaging heavy downpours. A recent example is the rainfall event that occurred in Dakar in the morning of August 26, 2012, causing the largest flood over the last twenty years in the city. An amount of 160 $mm$ was recorded at the Dakar-Yoff station which is large but not a historical record at this station. In fact, this event was exceptional because of its intensities at short durations (54 mm were recorded in 15 $minutes$ and 144 $mm$ in 50 $minutes$) exceeding by

far the previous records in Dakar-Yoff. Such rainfall intensities and their associated disasters justify the importance of better documenting extreme rainfall distributions at short time scales.

## 2.2 Rainfall Data

The archives of climate/hydrological services of West African countries sometimes contain large amount of sub-daily rainfall records. However these records are most of the time stored in paper strip chart formats requiring a tedious task of digitization

for using them in numerical applications.

The present study has been made possible thanks to an important work of analyzing and digitizing rain-gauge charts carried out for the main synoptic stations of Senegal. This process was undertaken by the laboratory of hydro-morphology of the Geography Department of the University Cheikh Anta Diop of Dakar (UCAD) in collaboration with the National Agency of Civil Aviation and Meteorology (ANACIM) who provided the rainfall paper charts.

Senegalese synoptic stations are equipped with tipping bucket rain-gauges; the receiving ring is 400 $cm^2$ and a bucket corresponds to 0.5 mm of rain. The roll rotation is daily. The chart analysis has been performed with the software "Pluvio" developed by (Vauchel, 1992) allowing the computation of 5-minute time-step digitized rainfall series from the paper diagrams. It is a long and laborious task, which has the advantage of allowing a careful "chart by chart" checking of the quality of





the records before digitization. For more information on the digitization process, the reader may refer to the publications of Laaroubi (2007) and Bodian et al. (2016).

A total number of 23 tipping bucket rain-gauges was analysed, with data going back to 1955 for the oldest and to 2005 for the most recent. As the assessment of extreme rainfall distributions is known for being much sensitive to sampling effect and

erroneous data (Blanchet et al., 2009; Panthou et al., 2012, 2014b), a particular attention was paid to check and select the most appropriate series.

The data selection had to conciliate two constraints: (i) keeping the data set as large as possible and (ii) eliminating series that contain too much missing data.

The procedure for classifying one year-station as valid or not is the following: (i) first, the annual number of 5-min data and

the annual amount of rain are computed, (ii) the mean inter-annual values of these two statistics are computed on the whole series, (iii) a year is classified as valid if either the number of 5-min rain data or the amount of rainfall is comprised between 1 / 2.5 and 2.5 times their mean inter-annual values, (iv) other years are classified as missing and removed from the whole series. Since missing years influence the mean inter-annual values, step (ii), (iii) and (iv) are repeated until all remaining years are classified as valid (note that, in fact, no station-year had to be excluded after the initial step). All valid years for all series are

plotted on Figure 3. In order to keep the IDF fitting robust, only series with at least 10 years of valid data have been used. This led us to retain 14 stations with record length varying from 10 (Fatick station) to 44 years (Ziguinchor station) with a median of 28 years. This dataset has the advantage of fairly covering the whole country but as the length of the series varies, the quality of the IDF estimates might differ from one station to another. This effect will be more precisely analysed in section 5.2.1.

## 3 Theoretical background

### 3.1 General IDF formulations

#### 3.1.1 Empirical IDF formulations

Intensity-Duration-Frequency (IDF) curves provide estimates of rainfall intensity for a range of durations $\{D\}$ and for several frequencies of occurrence (usually expressed as a return period $T$). Each curve corresponds to the evolution of a return level ($i_T$) as a function of rainfall duration $D$. Historically, several empirical formulations of IDF curves have been proposed. All

can be described by the following general equation (Koutsoyiannis et al., 1998):

$$i_T(D) = w(T) \times [D + \theta(T)]^{\eta(T)} \tag{1}$$

where $w$, $\theta$ and $\eta$ are parameters to be calibrated from rainfall observations.

#### 3.1.2 Koutsoyiannis scaling relationship

Koutsoyiannis et al. (1998) have demonstrated that the empirical formulations (Equation 1) can be expressed as :

$$i_T(D) = a(T) \times b_{\text{Koutso}}(D) \tag{2}$$





where $b_{\text{Koutso}}(D)$ is the scaling function:

$$b_{\text{Koutso}}(D) = (D + \theta)^{\eta} \qquad (3)$$

The advantage of Equation 2 as compared to Equation 1, is to separate the dependency on $T$ (return period) from the dependency on $D$ (duration): $a(T)$ only depends on $T$, and $b_{\text{Koutso}}(D)$ only depends on $D$. A consequence is that for the particular case of $D_0 = 1 - \theta$

$$i_T(D_0) = a(T) \qquad (4)$$

Then, it becomes a classical frequency analysis of the random variable $I(D_0)$ to estimate the return levels $i_T(D_0)$ - i.e. study $\mathbb{P}[I(D_0) \leq i(D_0)]$. Then, Equation 2 can be reformulated as an equality of distribution of random variables $I$:

$$I(D) \overset{d}{=} I(D_0) \times b_{\text{Koutso}}(D) \qquad (5)$$

### 3.1.3 Simple scaling relationship

In the particular case of $\theta = 0$, Equation 5 becomes:

$$I(D) \overset{d}{=} I(D_0) \times b_{\text{SiSca}}(D) \qquad (6)$$

$$b_{\text{SiSca}}(D) = D^{\eta} \qquad (7)$$

where $b_{\text{SiSca}}(D)$ is a simple scaling formulation of $b$.

### 3.2 IDF scaling formulations in the frame of the Extreme Value Theory

In the scaling approach described above, the estimation of rainfall return levels requires a statistical model of rainfall intensity distribution since equations 5 and 6 take the form of an equality of distributions. The extreme value theory (Coles, 2001) is the most commonly used framework for deriving these models.

### 3.2.1 Block maxima framework in Extreme Value Theory

The Extreme Value Theory (EVT) proposes two methods to extract samples of extreme values from a time series (Coles, 2001): the Block Maxima Analysis (BMA) which consists of defining blocks of equal lengths (often one year in hydrology) and extracting the maximum value within each block; the Peak Over Threshold (POT) which consists of extracting all the values exceeding a given threshold.

Compared to BMA, the POT has the advantage of allowing the selection of more than one value per year, thus increasing the sample size used for inferring the model, but the choice of an appropriate threshold is often difficult (Frigessi et al., 2002). Here the BMA approach was preferred as it is more straightforwardly implementable.


In BMA, when the block is large enough (which is ensured for annual maxima), the Extreme Value Theory states that the Generalized Extreme Value (GEV) distribution is the appropriate model for block maxima samples(Coles, 2001). The GEV distribution is fully described by three parameters, the location ($\mu$), the scale ($\sigma$), and the shape ($\xi$), which are respectively related to the position, the spread and the asymptotic behaviour of the tail of the distribution:

$$F_{GEV}(i;\mu,\sigma,\xi) = \exp\left\{-\left[1+\xi\left(\frac{i-\mu}{\sigma}\right)\right]^{-\frac{1}{\xi}}\right\} \quad \text{for } 1+\xi\left(\frac{i-\mu}{\sigma}\right) > 0 \tag{8}$$

A positive (negative) shape corresponds to a heavy-tailed (bounded) distribution. When $\xi$ is equal to 0, the GEV reduces to the Gumbel distribution (light-tailed distribution):

$$F_{GUM}(i;\mu,\sigma) = \exp\left\{-\exp\left[-\left(\frac{i-\mu}{\sigma}\right)\right]\right\} \tag{9}$$

### 3.2.2 GEV parameter formulation in a scaling framework

Menabde et al. (1999) have derived the equations merging the scaling formulations presented above (both $b_{\text{SiSca}}$ and $b_{\text{Koutso}}$) with the extreme value distribution (see also Panthou et al., 2014b; Blanchet et al., 2016). In this approach, the $I(D)$ samples are modeled by a GEV model for which the location and scale parameters are parameterized as a function of $D$ as follows:

$$I(D) \sim \quad \text{GEV}\{\mu(D);\sigma(D);\xi\} \tag{10a}$$

$$\mu(D) = \quad \mu_0 \times b(D) \tag{10b}$$

$$\sigma(D) = \quad \sigma_0 \times b(D) \tag{10c}$$

The return levels are easily obtained at all durations $D$ as:

$$i_T(D) = F_{GEV}^{-1}\left(D, 1-\frac{1}{T}\right) \tag{11}$$

This formulation is equivalent to the following:

$$i_T(D) = F_{GEV}^{-1}\left(D_0, 1-\frac{1}{T}\right) \times b(D) \tag{12}$$

With $D_0 = 1 - \theta$.

Note that equations 10 to 12 are valid for both $b_{\text{SiSca}}$ and $b_{\text{Koutso}}$.

## 4 Methodology: inference, evaluation and uncertainty of IDF models

In this study, two IDF models are compared: the IDF$_{\text{Koutso}}$ obtained from the Koutsoyiannis scaling $b_{\text{Koutso}}$ and the IDF$_{\text{SiSca}}$ obtained from the simple scaling $b_{\text{SiSca}}$. Both models describe the distribution of extreme rainfall intensities across duration but they differ in their formulation and in their number of parameters: IDF$_{\text{SiSca}}$ has four parameters $\{\mu_0, \sigma_0, \xi, \eta\}$ while



IDF$_\text{Koutso}$ has five parameters $\{\mu_0, \sigma_0, \xi, \eta, \theta\}$. The BMA samples from which the two scaling models are inferred and evaluated are built by using 1-year block lengths, in order to ensure the independency between the elements of the sample. At each station, the extreme rainfall sample thus consists of annual maximum intensities $i(D)$ with $D$ ranging from 1h to 24h : $\{1, 2, 3, 4, 6, 8, 10, 12, 15, 18, 24\}$h. Note that, for each duration $D$, a rolling mean of length $D$ is applied to the 5 minute rainfall

series before extracting the maxima. This ensures that the extracted maxima are not under-estimated (which is the case when using a fixed window).

## 4.1 IDF model inference

Different fitting methods have been tested to adjust the IDF model parameters to rainfall data: one of them (the two-step method) is applicable to both IDF$_\text{Koutso}$ and IDF$_\text{SiSca}$ models.

Note that two other methods specifically dedicated to the IDF$_\text{SiSca}$ model have also been tested: one based on the moment scaling function (as in e.g. Borga et al., 2005; Nhat et al.; Panthou et al., 2014b), and one based on the global maximum likelihood estimation (as in Blanchet et al., 2016). As they did not perform better than the two-step method, they are not presented here.

The fitting of the scaling $b(D)$ is based on the equality of distribution given in Equation 5 for IDF$_\text{Koutso}$ and Equation 6 for

IDF$_\text{SiSca}$. If these equations hold, the scaled random variables $I(D)/b(D)$ have for all durations $D$ the same distribution as the random variable $I(D_0)$. This means that the observed scaled samples $i(D)/b(D)$ have similar statistical properties for each duration $D$. Based on this property, the parameters of $b(D)$ are calibrated in order to minimize a statistical distance between the different scaled samples $i(D)/b(D)$. As suggested by Koutsoyiannis et al. (1998), the difference in medians computed by the Kruskal-Wallis statistic applied on multi-samples (Kruskal and Wallis, 1952) was chosen to characterize the distance between

the scaled samples $i(D)/b(D)$.

Once the scale relationship is identified, scaled samples $i(D)/b(D)$ are computed and pooled in a single sample since they are expected to follow the same GEV distribution (see Equation 12). The GEV parameters are estimated on this aggregated scaled sample by using the L-Moments method. This method was retained as it is more suitable for small samples (Hosking and Wallis, 1997) than the Maximum Likelihood Estimation algorithm which sometimes fails in optimizing the likelihood for

too small samples.

## 4.2 Models evaluation and selection

With the aim of selecting the best IDF model from the two compared IDF formulations (IDF$_\text{Koutso}$ and IDF$_\text{SiSca}$), a process of model evaluation and comparison is proposed here by looking at both their flexibility (the models are fitted on a calibration sample) and their robustness (the models are fitted in a predictive mode on samples not used for calibration).

The flexibility characterizes the capacity of a model to fit the observed data which are used to calibrate its parameters. To that purpose, the IDF models are fitted at each station; then different scores are computed to assess the fitting performances.

The robustness, on the other hand, aims at evaluating whether the flexibility is not overstretched due to the model having too many parameters with respect to the number of observations. As the two models tested here have a different number of



parameters (4 for IDF$_{SiSca}$ , 5 for IDF$_{Koutso}$ ), there is a particular interest at comparing between models how the goodness of fit is degraded when shifting from the calibration mode to the predictive mode. The predictive capacity of the IDF models is assessed by using a classical calibration/validation process. At each station, a subset of data is used to fit the IDF model; a second independent subset is used to validate it. The same scores as in the calibration mode are computed for the validation

subset. Rather than using two consecutive sub-periods, one for the calibration sample and one for the predictive sample, a year to year separation was used to build the two subsets. This limits the risk of obtaining samples made of years belonging predominantly to a dry period or to a wet period.

The flexibility and the predictive capacity of the IDF models are quantified based on two types of scores: global and quantile-quantile.

The two global scores are the statistics returned by two goodness of fit (GOF) tests: Kolmogorov-Smirnov (KS) and Anderson-Darling (AD). Each test computes a statistic based on the differences between a theoretical Cumulative Distribution Function (CDF) and the empirical CDF. The null hypothesis is that the sample is drawn from the fitted model. The test returns also the corresponding p-value (error of first kind). The p-value is used as an acceptation/rejection criterion by fixing a threshold (here 1%, 5%, and 10%). These tests and p-values were computed for each rainfall duration at each station.

GOF tests allow evaluating the whole distribution but do not guarantee that all quantiles are correctly estimated. Thus, as a complement, quantile based scores are also computed. They characterize the relationship between theoretical (obtained from the fitted CDF) and empirical quantiles (obtained from the empirical CDF). The root mean square error (RMSE), the mean error (ME), and the mean absolute error (MAE) quantile-based scores are computed. The full presentation of theses scores can be found in Panthou et al. (2012). A weighted version of these scores is also used in order to assign greater weight to unusual

quantiles, as proposed by Begueria and Vicente-Serrano (2006) and also presented in Panthou et al. (2012).

### 4.3   Uncertainty assessment

From a methodological point of view, the central contribution of this paper is its attempt at quantifying the uncertainty associated with IDF calculation in a scaling framework. This involves two distinct aspects. One is the uncertainty linked to the estimation of the scaling parameters. The other is the uncertainty linked to the inference of the GEV parameters. This second

component is especially important to consider when applying a scaling model to a location where daily rainfall series only are available, which is the ultimate purpose of regional IDF models. Indeed in some regional studies, the scaling parameters will have to be inferred from a very few stations where rainfall is recorded at subdaily time-steps; if they display variations in space, they then will have to be spatially interpolated so as to provide scaling parameter at any location of interest, notably at the location of daily rainfall stations. At these stations, the scaled GEV distribution is thus estimated from the daily observations only,

making the inference far less robust than when using a richer scaled sample obtained from observations ranging from one hour - or less - to one day.

Therefore, in the following, the uncertainty assessment at a given location will be addressed separately for the two situations: i) first when observations at this location are available over a whole range of time-steps; ii) secondly when only daily observations are available.





### 4.3.1 Uncertainty linked to the inference of the scaling model at locations with multi time-scale observations

Confidence intervals for IDF parameters and return levels are estimated using a non-parametric bootstrap (Efron and Tibshirani, 1994). For each station, it consists of fitting IDF curves to bootstrap samples $(i(D)_{boot})$ obtained from the original $i(D)$ samples. The entire process consists of three steps:

1. The vector of years is resampled with replacement (Monte Carlo resampling) until its length equals the length of the original vector.

2. Once a year $y$ is drawn in the bootstrap sample of years, the annual maximum for that year is retained for each duration $D$ in order to build the bootstrap sample of rainfall intensities $i(D)_{boot}$. This guarantees the coherence between the samples at different durations.

3. The IDF model is fitted on the bootstrap sample $i(D)_{boot}$.

4. The obtained parameters - $\{\mu_0, \sigma_0, \xi, \eta, \theta\}_{boot}$ for IDF$_{\text{Koutso}}$ and $\{\mu_0, \sigma_0, \xi, \eta\}_{boot}$ for IDF$_{\text{SiSca}}$ - and the associated return level $i_T(D)_{boot}$ are stored.

These four steps are repeated 1000 times leading to generate 1000 $i(D)_{boot}$ samples and obtained vectors of length 1000 for the different IDF parameters and for the different IDF return levels stored in step 4. Confidence intervals are computed on these
vectors. It is important to underline that these confidence intervals are a measure of the global uncertainty associated with the inference IDF model (uncertainty due to the inference of the scaling relationship and uncertainty generated by the inference of the parameters of the scaled GEV).

### 4.3.2 GEV versus donwscaling sources of uncertainty at locations where only daily observations are available

When only daily observations are available, the GEV parameters are inferred on the corresponding annual block maxima
sample of daily data, which contains far less information that the scaled samples used for fitting a scaled GEV when multi timescale observations are available. The GEV parameters for the sub-daily time-steps are then deduced from the daily GEV parameters using scaling parameters that must be inferred from nearby multi timescale observations. In some cases this might generate a significant departure from the GEV model that would have been fitted directly on the observations at the proper time-step if they were available. This effect is studied here by assuming that only the daily data were available for fitting the
GEV at our 14 stations and implementing the bootstrap approach in a way that allows separating the uncertainty linked to the GEV parameter inference and the uncertainty linked to the inference of the scaling parameters. This involves two independent bootstrap resampling processes.

The first consists in resampling $i(24h)$ based on 1000 bootstrap drawings and fitting 1000 GEV(24h) to these bootstrap samples. These 1000 GEV(24h) are then downscaled to a target duration $D$ using Equation 10, yielding 1000 different GEV($D$)
(practically only the results obtained for the 1-hour duration are presented here). The scaling parameters used to inform Equation 10 are those computed from the complete multi time-step data set as explained in section 4.1. This process yields a sample



of 1000 GEV at 1h duration - denoted $\{GEV(1h)\}_{GEV}$. The dispersion of these 1000 GEV(1h) is linked to the sole sampling effect underlying the adjustment of the initial GEV(24h), assuming the scaling parameters to be perfectly known.

In a parallel way the uncertainty associated with scaling is evaluated by generating 1000 downscaled samples from the reference GEV(24h) fitted on the original sample $i(24h)$ using the bootstrap procedure described in the previous section

(4.3.1). This produces a sample of 1000 GEV(1h) denoted $\{GEV(1h)\}_{Scal}$, whose internal dispersion is only influenced by the uncertainty in inferring the scaling parameters, assuming the reference GEV(24h) to be perfectly known.

## 5   Results

### 5.1   Model evaluation and selection

The results of model evaluation are presented in Figure 4, Figure 5 and Table 1. In these figures and table, the subscript a (resp.

b) relates to the calibration (resp. validation) results.

Figure 4 presents the GOF p-value of the KS test obtained for both models in calibration and validation mode at each station (the AD test gives similar results, not shown). In Figure 5, all stations are gathered in one single qq-plot from which global scores are computed. All global results (non-weighted and weighted qq-scores) are reported in Table 1.

### 5.1.1   Flexibility and Robustness

Figure 4a shows that for all stations and durations the KS p-values are higher than 10% (i.e. the risk of being wrong by rejecting the null hypothesis "observations are drawn from the models" is greater than 10%). This means that both IDF models fit the observed data with a reasonable level of confidence in calibration and have thus good flexibility skills. The global scores reported in Figure 5a and Table 1 show that in calibration, $IDF_{Koutso}$ slightly outperforms $IDF_{SiSca}$ . This result was expected as $IDF_{Koutso}$ has an additional degree of freedom ($\theta$ parameter) compared to $IDF_{SiSca}$ .

As regarding the validation mode four stations display p-values below 10% at almost each duration (Figure 4b); globally, both models display a similar number of occurrences of p-values below 10% (37 for $IDF_{SiSca}$ and 35 for $IDF_{Koutso}$ ) as well as below 5% (21 for $IDF_{SiSca}$ and 20 for $IDF_{Koutso}$ ) and below 1% (1 for $IDF_{SiSca}$ and 2 for $IDF_{Koutso}$ ).

The global qq-plots in Figure 5 and the statistics summarized in Table 1 confirm that the two IDF models perform very similarly in validation. $IDF_{SiSca}$ has slightly smaller biases (mean errors) while RMSE and MAE are slightly better for $IDF_{Koutso}$

25   .

### 5.1.2   Model selection

In addition to performing closely to each other in both calibration and validation modes, the two models yield very similar parameters and return levels, as may be seen from Figure 6. It is worth noting that the fitted values of the additional parameter $\theta$ of the $IDF_{Koutso}$ model range from -0.02 to 0.39, which is relatively close to zero as compared to the [1h–24h] range of





durations considered here. This means that the model is de facto very close to the IDF$_{SiSca}$ model, which is a simplification of the IDF$_{Koutso}$ model assuming $\theta$ being equal to zero.

Consequently, while there is no factual reason for considering one of the models to be better than the other, the IDF$_{SiSca}$ model will be retained, according to the following considerations:

1. it is more parsimonious with no clear advantage brought by the fifth parameter of the IDF$_{Koutso}$ model;

2. it is easier to implement, especially in a perspective of regional studies involving the mapping of the scaling parameters;

3. there is a straightforward link between the formulation of the IDF$_{SiSca}$ model and that of the Montana formula (see appendix) commonly used in national or regional agencies; this makes the formulation of the final IDF product easier to grasp by end-users, thus facilitating its adoption and use.

## 5.2 Assessing the Uncertainties

### 5.2.1 Global uncertainty when multi time-scale samples are available

The bootstrap approach presented in section 4.3.1 yields confidence intervals representing the global uncertainty linked to sampling in a situation where several samples at different time-steps are available at the same location. More precisely, it makes a Monte-Carlo exploration of how the aggregated scaled sample built from the multi time-scale initial samples may vary
depending on the random variations of each initial sample. The results are presented in Figure 7 for four major cities spread over Senegal. Three of them have all their GOF p-values above 0.1 in both calibration and validation modes (Figure 4) while the fourth (Dakar) has its GOF p-values mostly below 0.1 in validation mode, a few of them being even below 0.05 (meaning that, at that particular station, the model is less skilful).

The 90% confidence intervals of the IDF curves are displayed as coloured stripes in Figure 7. As intuitively expected, for a
given station, the higher are the return periods considered, the larger are the confidence intervals. Equally conform to knowledge and practice is the fact that, for a given parameter, the largest uncertainty intervals are usually obtained for the shortest series (Fatick, Podor, and Thies), while the longest series (Dakar-Yoff, Tambacounda, Kaolack, and Ziguinchor) display the narrowest intervals (Table 2). However this relation weakens when considering higher moments of return periods: the sample size explains 80% of the variance of the confidence interval width for $\mu$, 70% for $\sigma$, 55% for $\xi$ and 4% only for $i_{T=100}$. The presence of
very rare events in an observed sample is another factor widening the confidence intervals because some bootstrap samples will include these values, while others will not.

When comparing the confidence intervals computed for each parameter of the scaled GEV, it appears that their width is well correlated between $\mu$ and $\sigma$ ($r^2 = 0.82$) and much less so between $\mu$ or $\sigma$ and $\xi$ ($r^2 = 0.32$ between $\sigma$ and $\xi$). The widths of the confidence intervals are quite large for both $\sigma$ and $\xi$ which had to be expected since 8 stations out of 14 have a sample
size smaller than 30. The uncertainty on $\xi$ is a sensitive issue, since it involves that this parameter may take negative values, implying a bounded behaviour (Weibull domain of attraction), whereas a light (zero shape – Gumbel domain of attraction) or heavy (positive shape value – Frechet domain of attraction) behaviour is usually expected for rainfall extremes. It is however



slightly positive in average (+0.046) which tends to confirm the results obtained by Panthou et al. (2012, 2013, 2014b) in the central Sahel region, pointing to a dominantly heavy tail behaviour.

### 5.2.2 Scaling versus GEV related uncertainty when daily samples only are available

As previously explained, at stations where daily data only are available, the sub-daily GEV distributions have to be estimated

from this limited set of 24-hour values which significantly increases the uncertainty as may be seen from Figure 8. In this figure, the total uncertainty on the 1-hour GEV distribution is separated between the uncertainty linked to the initial fitting of the 24-hour distribution – GEV(24h) uncertainty – and the uncertainty generated by using the scaling relationships of equations 10b and 10c in order to downscale to 1-hour distribution GEV(1h) – scaling uncertainty. This decomposition is carried out by following the procedure presented in section 4.3.2. The results are given for the two longest series of our data set (Dakar Yoff,

38 years; Ziguinchor, 44 years), which happen to display two different behaviours. At Dakar Yoff, the GEV(24h) uncertainty becomes clearly larger than the scaling uncertainty from the 10-year return period onwards; at Ziguinchor, this occurs only from the 100-year return period onwards. Associated with this difference is the fact that the downscaled GEV model (dots in Figure 8) diverges from the reference scaled model (continuous line in Figure 8) for Dakar Yoff while they are almost identical for Ziguinchor. At Dakar, the width of the 90% confidence interval associated with the estimation of GEV(24h)

reaches 130 mm h$^{-1}$ for a return period of 500 years, against 30 mm h$^{-1}$ for the confidence interval associated with the scaling uncertainty. At Ziguinchor the values are respectively 50 mm h$^{-1}$ and 20 mm h$^{-1}$.

Figure 9 synthesizes the results obtained at all stations, basically confirming that the inference of the daily scale GEV(24h) is a far more important source of uncertainty than the inference of the scaling relationship, when it comes to estimate the GEV(1h). Figure 9 displays the minimum, mean, and maximum uncertainty spread obtained on the 14 stations for GEV(24h)

on the one hand (red) and the scaling relationship on the other (blue); the 50% shaded interval contains the 7 central values. In order to be able to compute these spreads, the values are expressed as a percentage of the rainfall value given by the GEV(1h) for each station at a given return level. It turns out that the spreads due to the GEV(24h) fit using daily samples are 3 to 4 times higher than those due to the scaling estimate for the 100-year return level and 5 times larger for the 500-year return level.

### 5.3 IDF products

### 5.3.1 IDF curves

A typical representation of of IDF curves is given in Figure 7. As a result of the IDF model formulations and the fitting on a unique scaled sample (for both IDF$_{Koutso}$ and IDF$_{SiSca}$ ), the return level curves are parallel (they do not cross) and the intensities decrease as the duration increases. The log-log linearity between return levels and durations comes from the simple scaling formulation (the curves would be bended but still parallel, for the IDF$_{Koutso}$ model). Rainfall return levels are of similar order

of magnitude for the four stations, even though a North-South gradient is apparent with rainfall intensities gradually increasing from Saint Louis to Dakar and from Dakar to Ziguinchor. At the 2-year return period, rainfall intensities vary from roughly 40 mm h$^{-1}$ (between 33 and 60 mm h$^{-1}$ when considering all 14 stations) for the 1h duration to approximately 3 mm h$^{-1}$





(between 2 and 5 mm h$^{-1}$) for the 24h duration. For any station, the return levels for the 10-year (resp. 100-year) return periods are approximately 1.5 (resp. 2) times higher than the 2-year return levels; these ratios hold at all time scales (from 1h to 24h) as a result of the log-log linearity of the intensity versus the duration. As already discussed in section 5.2, the novelty of these IDF curves is the fact that they are provided with their confidence intervals, allowing the user to get a representation of the uncertainty surrounding the estimated intensity return levels, which is linked to both the sample size and by the quality of the whole GEV&scaling model.

### 5.3.2 IDF mapping for Senegal

Maps of the 4 IDF parameters (GEV + scaling) over the whole Senegal are plotted in Figure 10. They have been produced by kriging the parameters inferred at each of our 14 stations. Two of these parameters ($\xi$ and $\eta$) are independent of the duration $D$, while $\mu$ and $\sigma$ are functions of $D$; these two parameters are thus mapped for the reference duration of 1h only. They both display a clear North-South increasing gradient, a feature already found by Panthou et al. (2012) for the Central Sahel: the location (resp. scale) parameter ranges from around 30 mm h$^{-1}$ (resp. 10 mm h$^{-1}$) in the North to around 50 mm h$^{-1}$ (resp. 15 mm h$^{-1}$) in the South. While there are different factors that may explain this gradient, it is clearly coherent with the similar gradient of the mean number of wet days (Figure 2) making the occurrence of a rainfall intensity less frequent in the North than in the South, simply because there are fewer rainfall events there (as evidenced for the whole region by Le Barbé et al., 2002).

As regarding the two non-duration dependent parameters the shape parameter $\xi$ does not display any clear spatial organization while the scaling parameter $\eta$ displays a South-West North-East gradient (with values ranging from -0.8 to -0.9). This suggests that, added to the latitudinal effect, the distance to the ocean might also influence the temporal structure of rainfall events. The values of the scaling parameter are very close to those observed by Panthou et al. (2014b) over the AMMA-CATCH Niger network located near Niamey.

The general pattern of the maps of 2-, 10- and 100-year return levels given in Figure 11 is almost totally driven by the North-South rainfall gradient for the 2- and 10-year return period. The pattern of the 100-year return period is a bit less regular, with the distance to the ocean seeming to play a role in the western part of the country and a higher patchiness that is certainly largely due to the sampling uncertainty at such a low frequency of occurrence.

## 6 Conclusions and discussion

### 6.1 Main results

This study of extreme rainfall over Senegal for durations ranging from 1h to 24h confirms previous research reporting that simple scaling seems to hold in tropical Africa for this range of time scales. The simplified GEV&scaling formulation proposed by Panthou et al. (2014b) using 4 parameters (3 for the GEV and 1 for the scaling) performs similarly to the 5-parameter formulation of Koutsoyiannis et al. (1998). This simplified formulation allows an easier study of the sampling uncertainties



associated with the inference of the 4 parameters, carried out by a bootstrap resampling in the observed sample of extreme rainfall at 14 stations. Thus in addition to establishing more solidly that scaling is an appropriate hypothesis for this region of the world, our study provides for the first time a comprehensive assessment of the different uncertainties affecting the IDF curves produced by the model (studies dealing with uncertainty focus on the whole IDF uncertainty, as Mélèse et al., 2017).

The key advantage of the GEV&scaling approach for computing IDF curves is twofold: 1) it ensures time-scale coherency (for the range of explored durations) when working at regional scale, thus allowing for a coherent spatial interpolation of the IDF model parameters over the region of interest; 2) it offers the possibility of deducing short durations GEV distributions at locations where 24h data only are available, thanks to this spatial interpolation. Both properties have been used in this paper. First, a one-out at a time simulation approach was used to explore the partition of the overall uncertainty between the GEV

inference uncertainty and the scaling model inference uncertainty. One important result in this respect is that the uncertainty produced by the inference of the GEV parameters is 3 to 4 times larger than the uncertainty associated with the inference of the scaling relationship. This means that the scaling relationship requires far less data to be inferred correctly than the GEV model. Secondly, maps of the 4 IDF model parameters and associated intensity return levels have been computed, allowing retrieving the general spatial pattern of these parameters over Senegal. The location ($\mu$) and scale ($\sigma$) parameters of the GEV

distribution, as well as the rainfall intensity levels for the 2-year and 10-year return periods, display a clear increasing gradient from North to South in line with the climatological gradient of the mean annual rainfall and of the occurrence of wet days. By contrast, for the temporal scaling parameter $\eta$ the increasing gradient is rather oriented from North-East to South-West, reflecting the influence of both the occurrence of wet days and of the distance to the ocean. The map of $\xi$ is somewhat patchy reflecting the fact that this parameter is usually difficult to estimate, but another important result of this study is that its average

value is slightly positive suggesting that the rainfall distribution is heavy tailed as often observed in several regions in the world (Koutsoyiannis, 2004b, a). Also worth noting is the fact that the value of $\eta$ is close to -1 (ranging from roughly -0.9 and -0.8) indicating a steep decrease of intensities as the duration increases. This is a common feature of short and intense rainfalls as those produced by convective storms. These values are comparable to those found by Mohymont et al. (2004) in the tropical area of Central Africa, and to those obtained in the Sahelian region of Niamey by Panthou et al. (2014b), close to -0.9 in both

cases; they are larger in absolute value than those found in mid-latitude regions, as already underlined by Van-De-Vyver and Demarée (2010).

A final consideration relates to the implementation of such IDF models in operational services. While the theoretical framework of coupling the GEV and scaling models might be considered as difficult to handle outside the world of academic research, their implementation to produce IDF curves is relatively easy, especially when using the simplified approach tested

here. This approach has the additional advantage of producing relationships between rainfall return levels formally equivalent to the so-called Montana relationship (see appendix), widely used in operational services, making easier the implementation and usage of our IDF model in meteorological/climatological services and hydrological agencies.





## 6.2 Points of discussion and perspectives

In the perspective of extending this work to other tropical regions of the world where subdaily rainfall data might be rare it remains to explore the effect of using a fixed window to extract the daily rainfall annual maxima, while a moving window was used for all durations (including 24h) in this study. As a matter of fact daily records of rainfall are carried out at a given hour of the day (usually 6:00 GMT or local time), producing smaller totals than when a mobile window is used to extract the daily rainfall maximum maximorum of a given year (or month). Since the scaling relationships that are used to deduced subdaily statistics from these fixed-window 24-hour maxima are tuned on multi-temporal maxima extracted with mobile windows, there is a potential underestimation bias of the subdaily statistics inferred at 24-hour stations that deserves to be studied.

Another critical question relates to using statistical inferences presupposing time stationarity in a context of a changing climate. Warming is already attested in the Sahel and is bound to increase involving possible changing annual rainfall patterns induced by changes in the positioning of the Bermuda-Azores High and of the Saharian Heat Low. Indeed, rainfall intensification in this region has already been reported by Panthou et al. (2014a) and by Taylor et al. (2017), likely in connection with an average regional warming of about 0.18K/decade over the past 60 years. While dealing with this question was far beyond the scope of this paper, it is a major challenge for both end-users and researchers. It requires developing non-stationary IDF curves, one possibility in this respect being to use both long historical rainfall series and the information that can be extracted from future climate model projections (see e.g. Cheng and AghaKouchak, 2014).

At the same time it is important to underline that stationarity is an elusive concept whose reality is never guaranteed in Nature, even without climate change. The Sahelian rainfall regime, for instance is known for its strong decadal variability (Le Barbé et al., 2002) with potentially great impacts on most extreme rainfall events (Panthou et al., 2013). The use of long rainfall series (multi-decadal) to fit IDF curves can thus reduce the sampling effects and reduce the IDF uncertainties but they can also introduce some hidden biases linked to this decadal-scale non-stationarity. This happened with the dams built on the Volta river in the seventies, and dimensioned based on the rainfall information of the previous three decades, two of which being abnormally wet. The reservoirs never filled up in the eighties and nineties. Therefore, while IDF curves are intended to be disseminated to a large community of end-users, they must be warned that they are nothing else than a decision making supporting tool to be used with care and to be updated regularly.

## Appendix A: Simple scaling IDF to Montana IDF

The IDF Montana formulation is as follows:

$$i_T(D) = a(T) \times D^{b_m} \tag{A1}$$





The underscript $_m$ is used to differentiate with the scaling expression $b$ in the main paper ($_m$ stands for Montana). In our case, the scaling function is the simple scaling (Equation 12), thus Equation A1 becomes:

$$i_T(D) = F_{GEV}^{-1}\left(D_0, 1 - \frac{1}{T}\right) \times D^\eta \tag{A2}$$

The two Montana parameters $a$ and $b_m$ can be derived by using the equality between the two formulations:

$$a(T) = F_{GEV}^{-1}\left(D_0, 1 - \frac{1}{T}\right) \tag{A3}$$

$$b_m(T) = \eta \tag{A4}$$

Note that when the simple scaling is verified then: (i) $D_0$ is equal to 1, and depend only on the unit chosen to expressed the intensity of rainfall; and (ii) the assumption on the dependence of $b_m$ on the return period $T$ in the Montana formulation is not more valid ($b_m$ is equal to $\eta$ for all return periods).

10 *Acknowledgements.* The research leading to these results received funding from the UK's National Environment Research Council (NERC) / Department for International Development (DFID) Future Climate For Africa programme, under the AMMA-2050 project (grant numbers NE/M020428/1, NE/M019969/1, NE/M019950/1, NE/M020126/1 and NE/M019934/1)



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





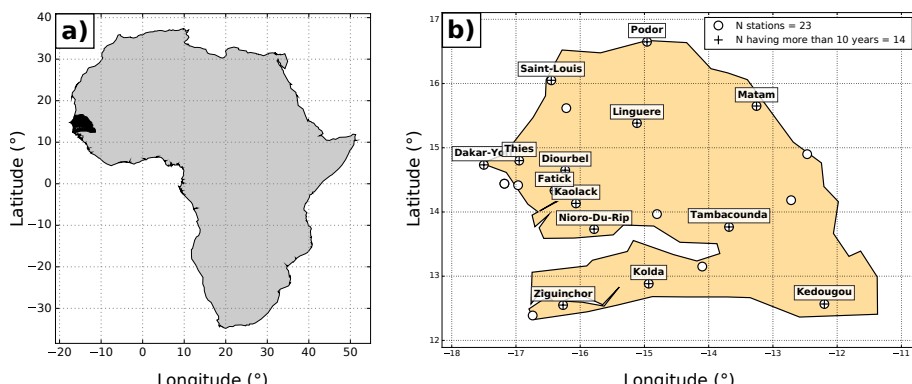

**Figure 1.** The Senegal localisation is represented by the black polygon (a). A zoom over Senegal (b): white circles represent tipping bucket rain-gauges (a cross and a label are added for stations having more than 10 years of valid data).

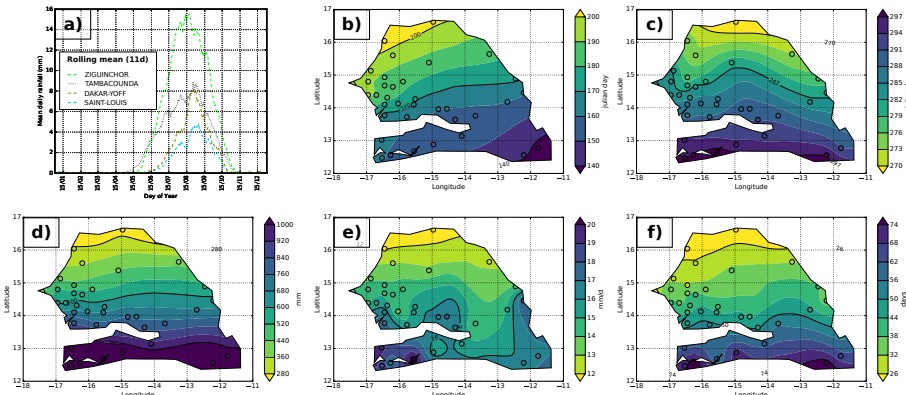

**Figure 2.** Rainfall regime statistics obtained from daily rain-gauges over the 1950-2015 period. Top panel : Mean seasonal cycle at four stations (a); start (b) and end (c) of the rainy season. On bottom panel: mean annual rainfall (d), mean intensity of wet days (e), mean number of wet days (f).





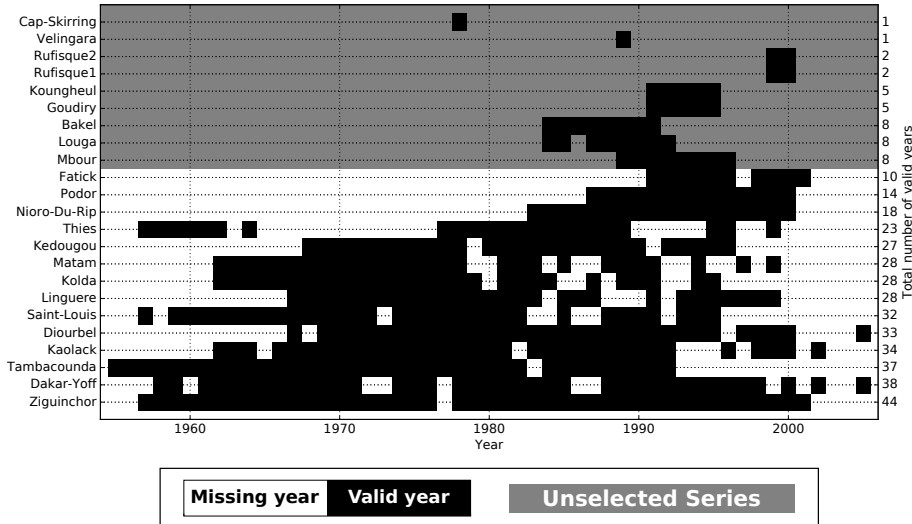

**Figure 3.** Network operation. The valid years are represented by black squares. The total number of valid years is displayed for each stations on right y-axis.





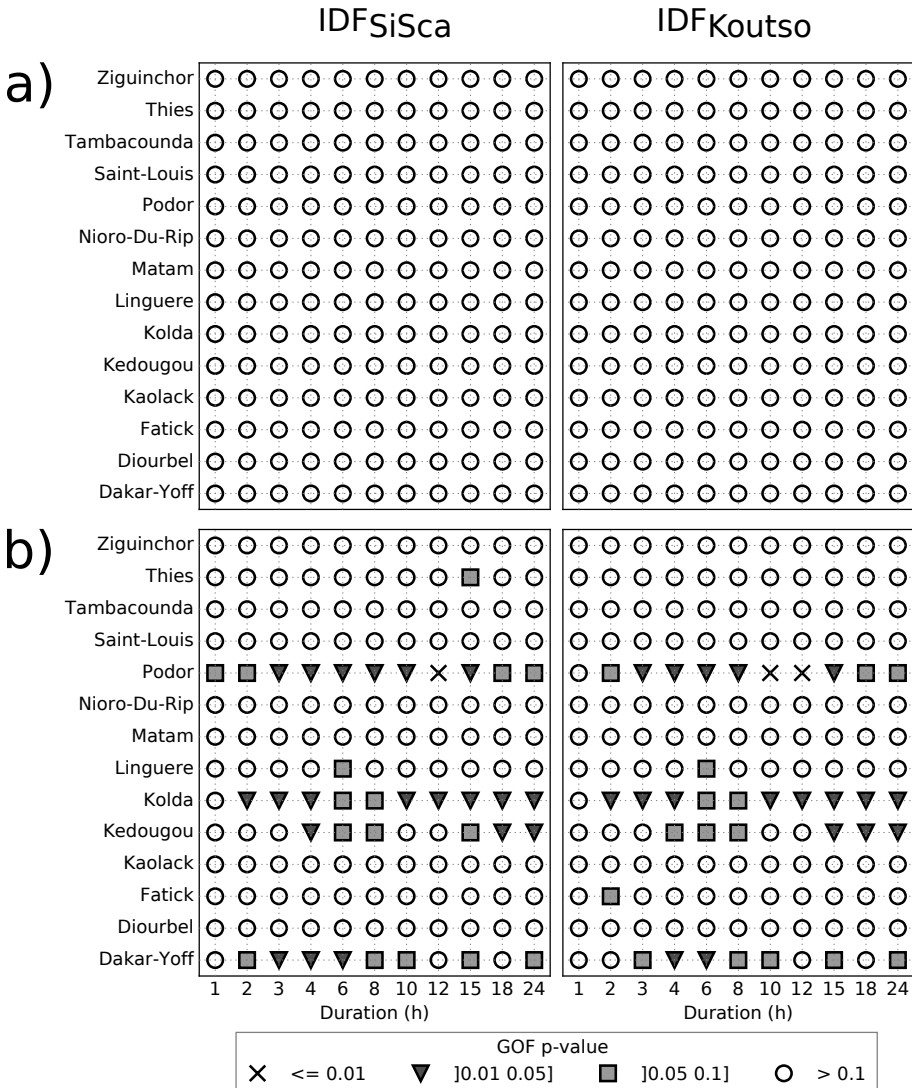

**Figure 4.** Kolmogorov-Smirnov GOF p-values for both IDF models : a) in calibration mode; b) in validation mode; left panel for IDF$_{SiSca}$ ; right panel for IDF$_{Koutso}$ .



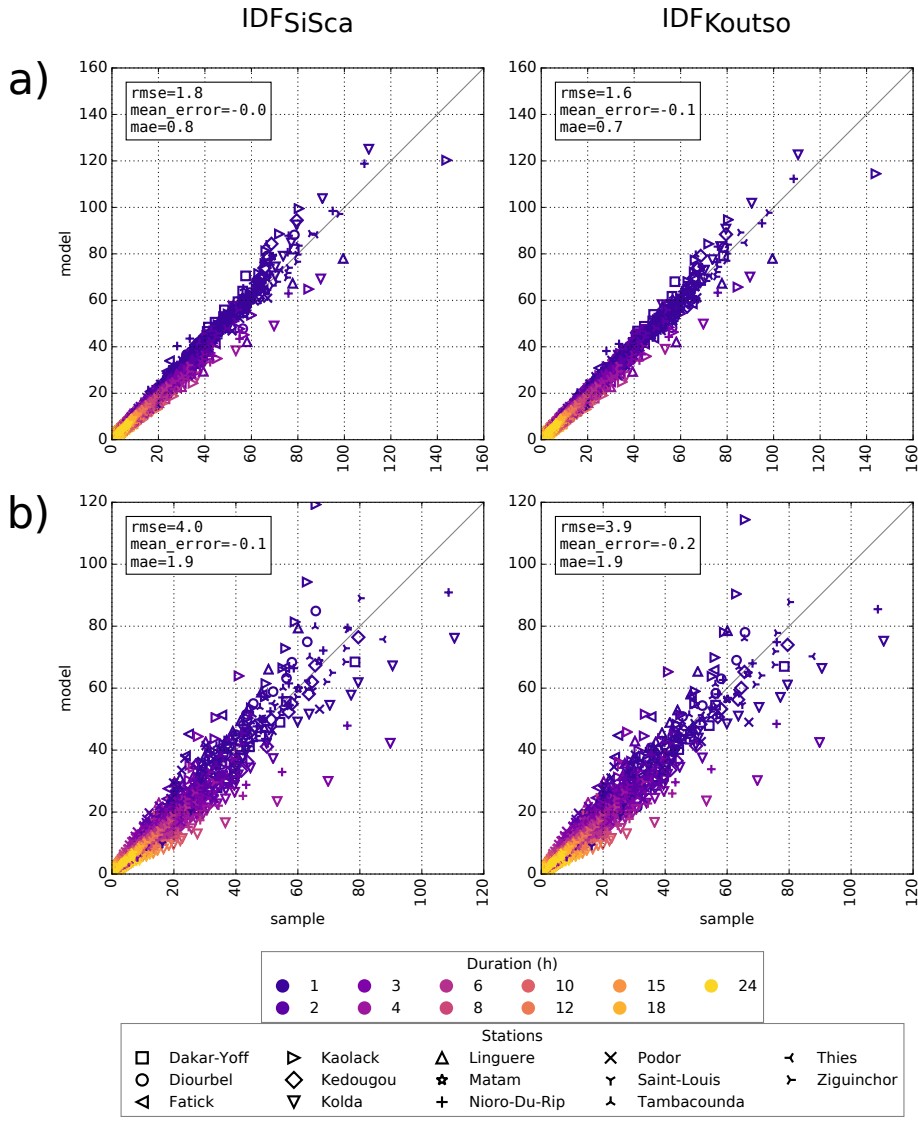

**Figure 5.** Quantile-quantile plots for both IDF models for the different duration and for all stations (global scores in the legend): a) in calibration mode; b) in validation mode; left panel for IDF$_{\mathrm{SiSca}}$ ; right panel for IDF$_{\mathrm{Koutso}}$ . The x-axis and y-axis units are $\mathrm{mm\,h^{-1}}$.



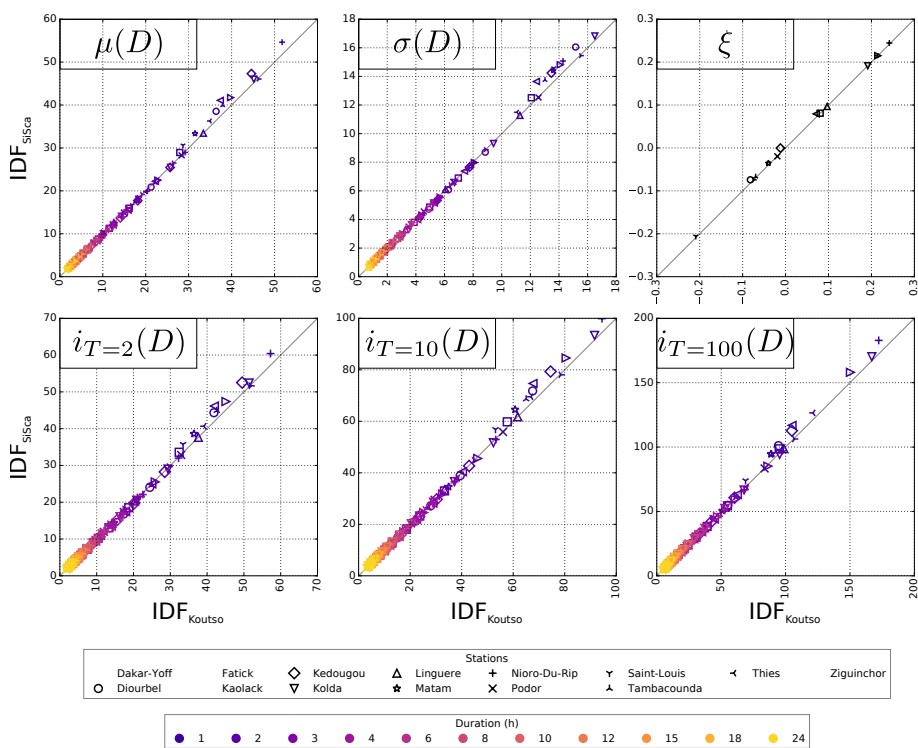

**Figure 6.** Comparison between IDF$_{\text{SiSca}}$ and IDF$_{\text{Koutso}}$ in calibration mode. On top panel : location parameter – $\mu(D)$ [mm h$^{-1}$] ; scale parameter – $\sigma(D)$ [mm h$^{-1}$]; and shape parameter – $\xi$ [$-$]. Return levels – $i_T$ [mm h$^{-1}$] – obtained for different return periods $T$ ranging from 2 to 100 years displayed in bottom panel.



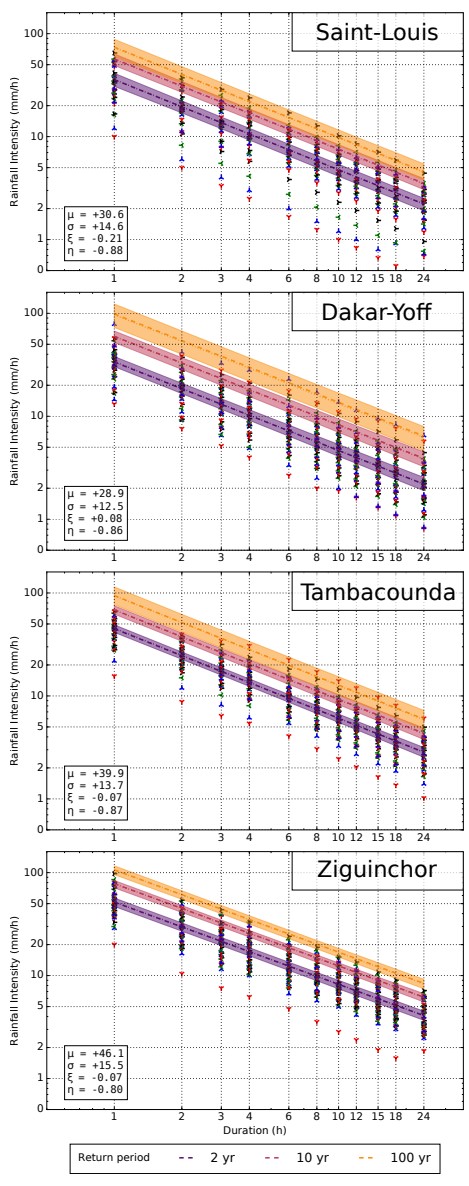

**Figure 7.** IDF$_{\text{SiSca}}$ return levels plots obtained at four emblematic stations over Senegal (from top to bottom : Saint-Louis, Dakar, Tambacounda and Ziguinchor). Shaded area represents the 90% confidence interval. The markers represent the observed annual maxima intensities $i(D)$.

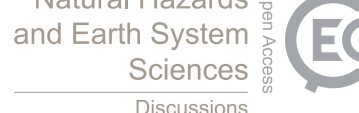



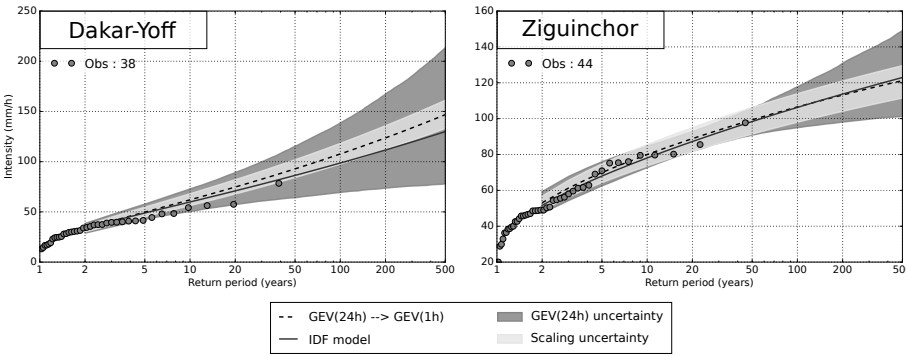

**Figure 8.** Return Level Plot for the 1h duration - GEV(1h) - at two emblematic stations over Senegal : Dakar and Ziguinchor. The estimates come from the GEV estimates at the 24h duration - GEV(24h) - downscaled to the 1h duration (dashed line). Shaded areas represent the 90% confidence interval of uncertainty due to the fitting of GEV(24h) and of uncertainty due to the downscaling. Obtained GEV(1h) from the fitted IDF at these stations are also displayed (solid line). The circles represent the observed annual maxima intensities $i(D)$.

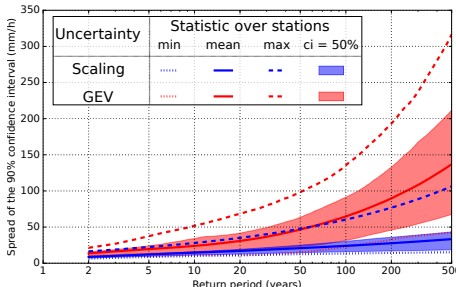

**Figure 9.** Evolution of the spread of the 90% confidence interval of return levels depending on the return period. The red color is for the spread due to the uncertainty of GEV(24h) fitting, while the blue color is for the spread due to the uncertainty of the scaling. All stations have been gathered. The mean, min, max, and 50% confidence interval of the spread obtained at the different stations are also shown.



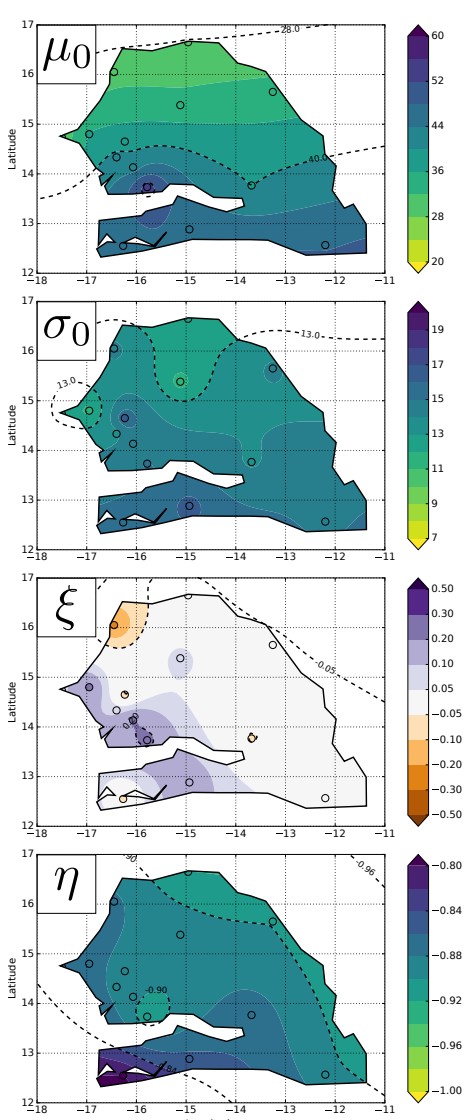

**Figure 10.** Maps of IDF$_{\mathrm{SiSca}}$ parameters : $\mu$ [mm h$^{-1}$], $\sigma$ [mm h$^{-1}$], $\xi$ [$-$], and $\eta$ [$-$].



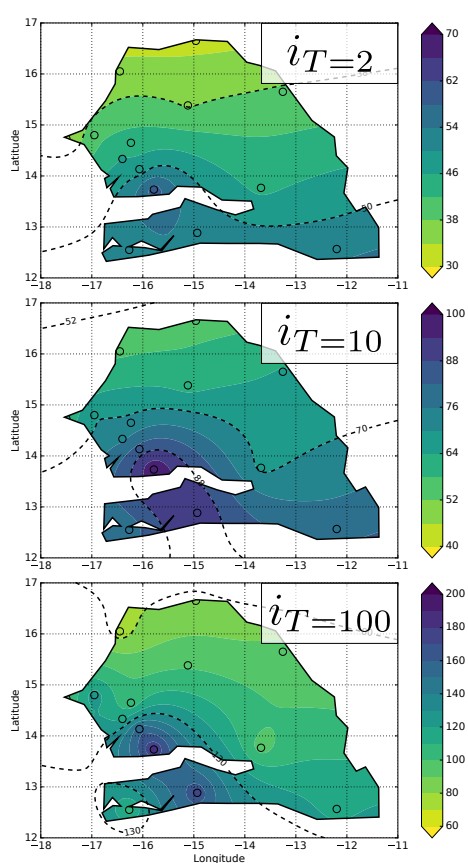

**Figure 11.** Maps of return levels intensities (from IDF$_{\text{SiSca}}$) $i_T$ [mm h$^{-1}$] for different return periods ($T$=2, 10, and 100 years from top to bottom) for the reference duration (1h).



**Table 1.** Global quantile-quantile scores results for the different IDF models: a) calibration mode; b) validation mode. All scores are expressed in $\mathrm{mm\ h^{-1}}$.

|  | rmse(classic) | rmse(weighted) | mean error(classic) | mean error(weighted) | mae(classic) | mae(weighted) |
|---|---|---|---|---|---|---|
| a) |  |  |  |  |  |  |
| IDF$_{\text{Koutso}}$ | 1.60 | 8.61 | -0.12 | -0.78 | 0.72 | 1.61 |
| IDF$_{\text{SiSca}}$ | 1.83 | 9.17 | -0.05 | -0.67 | 0.81 | 1.79 |
| b) |  |  |  |  |  |  |
| IDF$_{\text{Koutso}}$ | 3.87 | 12.13 | -0.17 | -0.71 | 1.91 | 3.12 |
| IDF$_{\text{SiSca}}$ | 3.96 | 12.40 | -0.10 | -0.60 | 1.94 | 3.15 |

**Table 2.** IDF$_{\text{SiSca}}$ fitted parameter values and 90% confidence interval estimated by bootstrap (in brackets).

| Unit | N # | $\mu$ mm h$^{-1}$ | $\sigma$ mm h$^{-1}$ | $\xi$ - | $\eta$ - | $i_{T=2}(D=1)$ mm h$^{-1}$ | $i_{T=10}(D=1)$ mm h$^{-1}$ | $i_{T=100}(D=1)$ mm h$^{-1}$ |
|---|---|---|---|---|---|---|---|---|
| Dakar-Yoff | 38 | 28.9 [26.1;32.9] | 12.5 [10.1;14.9] | 0.08 [-0.12;0.21] | -0.86 [-0.89;-0.83] | 34 [30;38] | 60 [52;67] | 99 [73;123] |
| Diourbel | 33 | 38.5 [33.3;44.7] | 16.1 [12.3;19.3] | -0.07 [-0.28;0.09] | -0.88 [-0.91;-0.86] | 44 [38;51] | 72 [63;80] | 101 [82;121] |
| Fatick | 10 | 41.1 [34.4;51.4] | 13.6 [6.3;18.9] | 0.08 [-0.31;0.34] | -0.89 [-0.93;-0.84] | 46 [37;58] | 75 [55;86] | 117 [79;141] |
| Kaolack | 34 | 41.7 [38.3;47.0] | 14.8 [11.3;19.2] | 0.21 [-0.07;0.36] | -0.89 [-0.92;-0.87] | 47 [43;54] | 85 [69;102] | 158 [99;225] |
| Kedougou | 27 | 47.3 [42.9;53.6] | 14.2 [10.0;17.9] | -0.00 [-0.24;0.18] | -0.89 [-0.92;-0.87] | 53 [47;60] | 79 [71;85] | 113 [99;123] |
| Kolda | 28 | 46.0 [42.1;52.7] | 16.8 [12.4;22.1] | 0.19 [-0.08;0.33] | -0.85 [-0.88;-0.82] | 52 [47;60] | 93 [76;110] | 170 [110;224] |
| Linguere | 28 | 33.4 [30.1;38.2] | 11.3 [8.8;14.0] | 0.10 [-0.20;0.24] | -0.89 [-0.92;-0.86] | 38 [34;43] | 62 [51;73] | 99 [66;132] |
| Matam | 28 | 33.4 [28.4;39.6] | 14.4 [10.5;18.0] | -0.04 [-0.23;0.12] | -0.90 [-0.93;-0.87] | 39 [33;46] | 65 [55;72] | 95 [79;107] |
| Nioro-Du-Rip | 18 | 54.6 [48.3;63.4] | 15.1 [10.0;23.7] | 0.24 [-0.05;0.35] | -0.92 [-0.95;-0.86] | 60 [53;71] | 100 [77;121] | 183 [107;221] |
| Podor | 14 | 28.3 [23.2;39.2] | 12.5 [6.9;17.7] | -0.02 [-0.44;0.26] | -0.92 [-0.98;-0.89] | 33 [26;45] | 56 [44;65] | 83 [68;97] |
| Saint-Louis | 32 | 30.6 [26.0;35.8] | 14.6 [11.3;17.5] | -0.21 [-0.40;-0.03] | -0.88 [-0.91;-0.84] | 36 [31;41] | 57 [49;64] | 74 [60;88] |
| Tambacounda | 37 | 39.9 [36.6;44.2] | 13.7 [11.1;16.0] | -0.07 [-0.27;0.09] | -0.87 [-0.90;-0.84] | 45 [41;49] | 69 [62;75] | 94 [77;114] |
| Thies | 23 | 36.3 [32.8;43.1] | 11.5 [7.7;16.4] | 0.22 [-0.01;0.34] | -0.88 [-0.92;-0.85] | 41 [36;49] | 70 [57;83] | 127 [90;156] |
| Ziguinchor | 44 | 46.1 [42.0;50.8] | 15.5 [12.2;18.4] | -0.07 [-0.21;0.05] | -0.80 [-0.82;-0.77] | 52 [47;57] | 78 [71;84] | 106 [96;116] |

N corresponds to the number of available years, thus the number of annual maxima.