# Peer review of "Intensity-Duration-Frequency (IDF) rainfall curves in Senegal"

_Natural Hazards and Earth System Sciences, 2017_

## Referee Comment (RC1) · A. Medard (Referee) · 15 Dec 2017

A. Medard (Referee)

agbazo.medard_noukpo@courrier.uqam.ca

First, I would like to congratulate the authors for choosing to work on this topic in one region of Africa. The paper focuses on the IDF curves; it is a societal topic of great importance for all countries of the world, but more specifically for those in Africa where the construction of road infrastructure, the forecast of floods and drought occupy much of their government's agenda. This article is well written and structured, and above all was carried out over long time series of rains that they treated well by a solid method. The Figures are clear and allow deducing the results. However, I have a very important question that is related to the methodology: I would like to ask the authors to explain the reasons for choosing the time scale interval from 1h to 24h only when they have a long database of durations D ranging from 5 minutes to 24 hours (5, 10, 15, 30, 60,

90, 120, 180, 240 min and 24 hours). I am not aware of the preliminary studies on the determination of scale invariance regimes in the rainfall time series in Senegal. Based for example on the work of (Ghanmi, 2015) for Tunisia and those of (Agbazo et al., 2016) for Benin, we know that from 5 minutes to 24 hours, there can be two regimes of invariance of scale and 1hour to 24 hours do not necessarily have one.

Thus, I would recommend to the authors if this is possible to make a study on the temporal regimes of scale invariance of their series to make sure that 1h to 24 hours is indeed a regime of scale invariance for Senegal. This would make the study more robust and complete.

I highly recommend the publication of this article.

---

## Referee Comment (RC2) · Anonymous Referee #2 · 2 Feb 2018

Page 3, line 11: As regards the IDF calculation for African countries the work of De Paola et al. (2014) should be considered. De Paola et al. (2014) also tried to asses how extreme rainfall will be modified in future climate performing analysis of observed data and future simulations in three African cities, Addis Ababa, Dar Es Salaam and Doula. De Paola et al. found a methodology for the evaluation of the IDF curve from daily rainfall data; to obtain duration shorter than 24 hours they applied two different models of disaggregation to the historical data available, later the IDF curves were obtained using the probability distribution of Gumbel. Finally, the same procedure was applied to rainfall projections over the time period 2010-2050 in order to estimate the influence of the climate change on the IDF curves. As regards the results of the climate model projections, they suggest an increase of rainfall in terms of frequency.

[Figure]

Page 3, line 13: The Extreme Value Distributions well interpret the maximum daily rainfall with reference to the city of Dar Es Salaam (See Cluva Chapter 2, Giugni et al. , The Impacts of Climate Change on African Cities, 2015); in particular, their work shows that the distribution of annual maxima is well modelled by the GEV distribution and that the shape parameter of the GEV distribution is essential to the determination of the characteristics of extreme value behaviour. Moreover, they showed that the estimation of GEV parameters by methods such as maximum likelihood can be unreliable in case of short rainfall records, but the estimation of the shape parameter done using the Bayesian method is more precise restricting the shape parameter to a physically reasonable range.

Page 7, lines 6-7: it is not really correct to state that the GEV distribution reduces to the Gumbel distribution when x is equal to zero; actually the GEV is not defined for x equal to zero, the GEV reduces to the Gumbel distribution when x tends to zero.

Page 8, lines 32-33: the definition of robustness is not clear since a robust statistic returns inferential results that are relatively insensitive to changes in the assumptions of the statistical model.

Page 12, lines 23-24: it is not clear from where we can deduce that considering higher moments of return periods the sample size explains 80% of the variance of the confidence interval width for $\mu$, 70% for s, 55% for x and 4% only for iT=100. Therefore this part should be better explained.

De Paola, F. Giugni, M., Topa, M.E. and Bucchignani E.,Intensity-Duration-Frequency (IDF) rainfall curves, for data series and climate projection in African cities SpringerPlus 2014, 3:133 doi:10.1186/2193-1801-3-133.

Giugni et al. The Impacts of Climate Change on African Cities, Chapter 2, Editors: Pauleit, S., Coly, A., Fohlmeister, S., Gasparini, P., Jørgensen, G., Kabisch, S., Kombe, W.J., Lindley, S., Simonis, I., Yeshitela, K. (Eds.), Urban Vulnerability and Climate Change in Africa, Springer, 2015.

---

## Author Comment (AC1) · 14 Mar 2018

Please find in attach: our response to reviewers's comments, a revised paper with tracks, and a revised paper.

Please also note the supplement to this comment:
https://www.nat-hazards-earth-syst-sci-discuss.net/nhess-2017-352/nhess-2017-352-AC1-supplement.zip

---

## Author Response (AR1)

**"Intensity-Duration-Frequency (IDF) rainfall curves in Senegal"**
**by Sane et al. (nhess-2017-352)**

**We first would like to thank the two reviewers for their comments. We have revised our manuscript and have responded point by point to each comment. Please note that we have also made some very minor revisions to improve the English thanks to an appreciative proofreading of an English-speaking (mother tongue) co-author. All co-authors have also checked the paper to eliminate any remaining typos.**

**We have also provided a revised paper version with tracked changes.**

**Reviewer #1 ()**

First, I would like to congratulate the authors for choosing to work on this topic in one region of Africa. The paper focuses on the IDF curves; it is a societal topic of great importance for all countries of the world, but more specifically for those in Africa where the construction of road infrastructure, the forecast of floods and drought occupy much of their government's agenda. This article is well written and structured, and above all was carried out over long time series of rains that they treated well by a solid method. The Figures are clear and allow deducing the results.

**Response:**

**Thank you for this general comment.**

However, I have a very important question that is related to the methodology: I would like to ask the authors to explain the reasons for choosing the time scale interval from 1h to 24h only when they have a long database of durations D ranging from 5 minutes to 24 hours (5, 10, 15, 30, 60, 90, 120, 180, 240 min and 24 hours). I am not aware of the preliminary studies on the determination of scale invariance regimes in the rainfall time series in Senegal. Based for example on the work of (Ghanmi, 2015) for Tunisia and those of (Agbazo et al.,2016) for Benin, we know that from 5 minutes to 24 hours, there can be two regimes of invariance of scale and 1hour to 24 hours do not necessarily have one.

Thus, I would recommend to the authors if this is possible to make a study on the temporal regimes of scale invariance of their series to make sure that 1h to 24 hours is indeed a regime of scale invariance for Senegal. This would make the study more robust and complete.

**Response:**

**Yes, we agree. The scaling regime question is both interesting and relevant. It takes two different aspects:**

- **First, as most studies do, one can focus on a range of durations defined arbitrarily, and then check whether or not a change in the temporal regimes of scale invariance is detected. This is the case in our study with a predefined range of time scales from 1h to 24h. This is justified by both operational and practical reasons: going below 1 hour seems difficult because extracting sub-hourly maxima from the raw 5min-series could potentially lead to a significant underestimation of the true maximum intensity since the 5-min window is fixed by construction (as opposed to the moving window procedure used for larger time steps); this sampling effect requires to accumulate several elementary time steps in order to be removed: working at one hour (accumulation of 12 elementary time steps) puts us on the safe side. For the upper end, the 24h bound has been chosen because it is the usual time sampling of national rain-gauge networks, providing in West African countries. This allows for evaluating the potential for a regionalization of IDFs over regions where only daily data are available. The hypothesis of a single temporal regime over the 1h-24h range is then checked by comparing between the Koutsoyiannis model (whose curvature might underline a**

transition between two temporal scaling regimes) and the simple scaling model (which implies a single temporal regime of scaling). The similar performances of these two models make it reasonable to assume the validity of a unique temporal regime of scaling over the 1h-24h range. The fact that all GOF for the simple scaling model (figure 4a) accept the null hypothesis is also in line with this hypothesis. In addition is worth noting that the literature dealing with this region or others tends to support the hypothesis of a simple scaling regime for this range of durations. Indeed, in the references given by the referee, there is no change of regime for this range of durations (1h – 24h): Ghanmi et al. (2016) find single temporal regime from 30 minutes up to 24h, while Agbazo et al. (2016) consider that there is a single *scaling regime from 5 min to 1440 min"*. And finally, the study of Panthou et al. 2014 also finds a single regime of scaling from 1h to 24h in another Sahelian region (Niger).

- This having been said, we recognize that a deeper investigation of scaling regimes for durations smaller than 1h would be valuable. However this would require better quality sub-hourly data. If such data were available, then we could develop a robust methodology to identify breaks in scaling regimes such as the one carried out by Innocenti et al. (2017) exploring temporal scaling regimes over North America for durations ranging from 15 minutes to 7 days. Their approach is interesting and certainly deserve attention, but it remains that there are very few papers dealing with sub-hourly scaling properties of rainfall, due to data limitation, such as is the case for our study.

In order to clarify the paper on the above discussed issue, three main modifications were carried out :

- **We recognize that our choice of duration range was not justified enough. We accordingly made the following changes:**

  **page 7 lines 3 to 6:** *"[…] At each station, the extreme rainfall sample thus consists of annual maximum intensities i(D) with D ranging from 1h to 24h : {1, 2, 3, 4, 6, 8, 10, 12, 15, 18, 24}h. […]"*

  **It now reads:** *"[…] At each station, the extreme rainfall sample thus consists of annual maximum intensities i(D) with D ranging from 1h to 24h: {1, 2, 3, 4, 6, 8, 10, 12, 15, 18, 24}h. The lower bound of this range (1h) was selected in order to limit the risk of under-estimating the true annual maximum intensity when evaluating at shorter durations (close to the 5-minute fixed window of the raw series). The upper bound of the range (24h) was chosen because it is a standard duration for hydrological applications and climate studies, but also because it is much more frequently recorded (by daily rain gauges).*

  *[…]"*

- **We also added in the methodology Section the difference in term of return level between the Koutsoyiannis scaling and the Simple scaling.**

  **Page 7 lines 22:** *"Note that equations 10 to 12 are valid for both b SiSca and b Koutso ."*

  **now reads:** *"Note that equations 10 to 12 are valid for both b SiSca and b Koutso. In log-log space, the IDFSiSca return levels have a linear shape, indicating a single temporal scaling regime, while those of IDFKoutso could present a more or less pronounced curvature, indicating a transition between two temporal scaling regimes."*

- **We also modified the conclusion in order to mention this question of temporal regime of scaling, by citing similar studies (Panthou et al. 2014, Agbazo et al. 2016, Ghanmi et al. 2016), and also the work of Innocenti et al. (2017) that provides some guidelines for further investigations .**

  **page 14, lines 28-30:**

  *"This study of extreme rainfall over Senegal for durations ranging from 1h to 24h confirms previous research reporting that simple scaling seems to hold in tropical Africa for this range of time scales. The simplified GEV&scaling formulation proposed by Panthou et al.*

*(2014b) using 4 parameters (3 for the GEV and 1 for the scaling) performs similarly to the 5-parameter formulation of Koutsoyiannis et al. (1998). This simplified formulation  [...]"*

**now reads**

*"This study of extreme rainfall over Senegal for durations ranging from 1h to 24h confirms previous research reporting that a single temporal regime of scale invariance (simple scaling) seems to hold in tropical Africa for this range of time scales (Panthou et al. 2014, Agbazo et al. 2016, Ghanmi et al. 2016). Whether this range could be extended to sub-hourly and/or sup-daily rainfall intensities is an open research question, out of the scope of this paper, but that can be apprehended using the recent methodology developed in Innocenti et al. (2017). The simplified GEV&scaling formulation proposed by Panthou et al. (2014b) with 4 parameters (3 for the GEV and 1 for the scaling) performs similarly to the 5-parameter formulation of Koutsoyiannis et al. (1998). This simplified formulation  [...]"*

I highly recommend the publication of this article.

**We thank the reviewer for its comments which – we hope – helped improve the clarity of the paper.**

**Reviewer #2 ()**

Page 3, line 11: As regards the IDF calculation for African countries the work of De Paola et al. (2014) should be considered. De Paola et al. (2014) also tried to assess how extreme rainfall will be modified in future climate performing analysis of observed data and future simulations in three African cities, Addis Ababa, Dar Es Salaam and Doula. De Paola et al. found a methodology for the evaluation of the IDF curve from daily rainfall data; to obtain duration shorter than 24 hours they applied two different models of disaggregation to the historical data available, later the IDF curves were obtained using the probability distribution of Gumbel. Finally, the same procedure was applied to rainfall projections over the time period 2010-2050 in order to estimate the influence of the climate change on the IDF curves. As regards the results of the climate model projections, they suggest an increase of rainfall in terms of frequency.

**Response.**

**Thanks for the reference. We added it page 3 line 11:**

*[…] at larger durations for a tropical climate. De Paola et al. (2014) have also inferred IDF curves from disaggregated daily rainfalls for three African cities (Addis Ababa, Ethiopia; Dar Es Salaam, Tanzania; and Douala, Cameroon).*

*More recently [...]*

Page 3, line 13: The Extreme Value Distributions well interpret the maximum daily rainfall with reference to the city of Dar Es Salaam (See Cluva Chapter 2, Giugni et al. , The Impacts of Climate Change on African Cities, 2015); in particular, their work shows that the distribution of annual maxima is well modelled by the GEV distribution and that the shape parameter of the GEV distribution is essential to the determination of the characteristics of extreme value behaviour. Moreover, they showed that the estimation of GEV parameters by methods such as maximum likelihood can be unreliable in case of short rainfall records, but the estimation of the shape parameter done using the Bayesian method is more precise restricting the shape parameter to a physically reasonable range.

**Response.**

**Thanks for the reference, we have thus added it with other references concerning the heavy tail of daily rainfalls.**

*"While Agbazo et al. (2016) assumed a Gumbel distribution of the annual maxima, Panthou et al. (2014b) used the approach in its broader formulation, showing that the annual maxima distribution was heavy-tailed (positive value of the shape parameter of the GEV). Indeed, such heavy-tailed behavior in daily rainfall samples is generally found: both in the African region (e.g. Panthou et al. 2012, Giugni et al. 2015) but also all around the world (e.g. Koutsoyiannis 2004, Papalexiou et al. 2013). "*

Page 7, lines 6-7: it is not really correct to state that the GEV distribution reduces to the Gumbel distribution when x is equal to zero; actually the GEV is not defined for x equal to zero, the GEV reduces to the Gumbel distribution when x tends to zero.

**Response.**

**Yes, we agree, the reviewer is absolutely right. The sentence now reads as follows: "** *when $\xi$ tends to 0, the GEV reduces to the Gumbel distribution* **"**

Page 8, lines 32-33: the definition of robustness is not clear since a robust statistic returns inferential results that are relatively insensitive to changes in the assumptions of the statistical model.

**Response.**

**We do not really agree with this remark. It depends on how the robustness is defined. Here the robustness refers to whether the IDF model is over-fitted or not.**

**Nonetheless, from the remark of the reviewer, we understand that this concept was not well explained in the paper. Thus, we reformulated the following paragraph:**

*"[…] to assess the fitting performances.*

*The robustness, on the other hand, aims at evaluating whether the flexibility is not overstretched due to the model having too many parameters with respect to the number of observations. As the two models tested here have a different number of parameters [...]"*

**now reads:**

*"[…] to assess the fitting performances.*

*The robustness, on the other hand, aims at evaluating whether or not the IDF model is too flexible due to the model having too many parameters with respect to the number of observations. It thus depends on the sensitivity of the IDF model parameters to sampling effects: the less the model parameters are sensitive to sampling effects, the more the model is robust. As the two models tested here have a different number of parameters [...]"*

Page 12, lines 23-24: it is not clear from where we can deduce that considering higher moments of return periods the sample size explains 80% of the variance of the confidence interval width for $\mu$, 70% for s, 55% for x and 4% only for iT=100. Therefore this part should be better explained.

**Response.**

**Thank for this relevant remark. In fact, there are two sources of confusion here:**

**First there was a typo:** *"of"* **rather than** *"or"* **in the first part of the mentioned sentence. We have thus corrected the sentence** *"However this relation weakens when considering higher moments of return periods:"* **; it now reads** *"However, this relation weakens when considering higher moments or higher return periods:".*

**Secondly, we agree with the reviewer that the second part of the sentence might be too short to clearly explain what we have in mind. It refers to the linear regression between the relative width of the confidence interval of a given parameter/return level and the number of available years.**

**Thus we have modified the second part of the sentence:**

*"the sample size explains 80% of the variance of the confidence interval width for $\mu$, 70% for $\sigma$, 55% for $\xi$ and 4% only for iT =100"*

**now reads**

*"the coefficients of correlation between the confidence interval width and the sample size (available number of years) are $r^2=0.80$ for $\mu$, $r^2=0.88$ for $\sigma$, $r^2=0.69$ for $\xi$, $r^2=0.55$ for iT =2 and iT =10, and $r^2=0.004$ only for iT =100."*

**We thank the reviewer for its comments which – we hope – helped improved the clarity of the paper.**

[revised manuscript text omitted]

N corresponds to the number of available years, thus the number of annual maxima.